# Immunological Fluid Biomarkers in Frontotemporal Dementia: A Systematic Review

**DOI:** 10.3390/biom15040473

**Published:** 2025-03-24

**Authors:** Philip Ahle Erichsen, Emil Elbæk Henriksen, Jørgen Erik Nielsen, Patrick Ejlerskov, Anja Hviid Simonsen, Anders Toft

**Affiliations:** Neurogenetics Clinic & Research Lab, Danish Dementia Research Centre, Rigshospitalet, 2100 Copenhagen, Denmark; philip.ahle.erichsen.02@regionh.dk (P.A.E.); emil.elbaek.henriksen@regionh.dk (E.E.H.); joergen.erik.nielsen.01@regionh.dk (J.E.N.); patrick.ejlerskov@regionh.dk (P.E.); anja.hviid.simonsen@regionh.dk (A.H.S.)

**Keywords:** immune, biomarkers, frontotemporal dementia, Alzheimer’s disease, amyotrophic lateral sclerosis

## Abstract

Dysregulated immune activation plays a key role in the pathogenesis of neurodegenerative diseases, including frontotemporal dementia (FTD). This study reviews immunological biomarkers associated with FTD and its subtypes. A systematic search of PubMed and Web of Science was conducted for studies published before 1 January 2025, focusing on immunological biomarkers in CSF or blood from FTD patients with comparisons to healthy or neurological controls. A total of 124 studies were included, involving 6686 FTD patients and 202 immune biomarkers. Key findings include elevated levels of GFAP and MCP1/CCL2 in both CSF and blood and consistently increased CHIT1 and YKL-40 in CSF. Complement proteins from the classical activation pathway emerged as promising targets. Distinct immune markers were found to differentiate FTD from Alzheimer’s disease (AD) and amyotrophic lateral sclerosis (ALS), with GFAP, SPARC, and SPP1 varying between FTD and AD and IL-15, HERV-K, NOD2, and CHIT1 differing between FTD and ALS. A few markers, such as Galectin-3 and PGRN, distinguished FTD subtypes. Enrichment analysis highlighted IL-10 signaling and immune cell chemotaxis as potential pathways for further exploration. This study provides an overview of immunological biomarkers in FTD, emphasizing those most relevant for future research on immune dysregulation in FTD pathogenesis.

## 1. Introduction

Frontotemporal dementia (FTD) refers to a group of neurodegenerative syndromes primarily characterized by changes in behavior, executive function, and language. The prevalence of FTD ranges from 1 to 460 per 100,000 in the general population [1]. Approximately 30% of FTD patients have a strong family history of dementia, with genetic variants in the *C9orf72*, *GRN*, and *MAPT* genes accounting for the majority of cases [2]. Clinically, FTD is categorized into three presentations: behavioral-variant FTD (bvFTD), featuring behavioral deficits and decline in executive functioning; semantic-variant primary progressive aphasia (svPPA), involving progressive loss of sematic knowledge and anomia; and non-fluent variant primary progressive aphasia (nfvPPA), marked by speech production deficits and agrammatism [3]. Over time, these phenotypes converge, with patients developing global cognitive decline and motor symptoms, frequently necessitating institutionalization. The average survival post-diagnosis ranges from 3 to 12 years, depending on the phenotype [4]. Currently, there are no disease-modifying treatments available to halt FTD progression, nor are there disease-specific biomarkers to aid in diagnosis, monitor disease progression, or evaluate treatment efficacy.

Abundant evidence implicates neuroinflammation and other maladaptive immune responses in FTD, including postmortem neuropathology, imaging studies, transgenic animal models, and genome-wide association studies [5]. Immune dysfunction may play both primary and secondary roles in disease progression. Primary immune dysfunction could emerge in the early stages of the disease, contributing to pathogenesis through mechanisms such as impaired microglial protein clearance or synaptic pruning [6]. In contrast, secondary immune responses may arise as a consequence of toxic protein deposition or neuronal damage, exacerbating neurodegeneration and disease severity [7]. This underscores the importance of further studies to explore the temporal dynamics and functional roles of immune alterations in FTD.

Considerable efforts have been made to identify immune-related biomarkers that can provide insights into FTD mechanisms, but their clinical utility remains limited [2]. The link between immune markers and FTD pathology involves both associative and potentially causal mechanisms. For instance, while elevated levels of specific cytokines and complement proteins may correlate with disease severity [8], it is unclear whether these immune alterations contribute to disease progression or are simply associated with its severity. Further validation through mechanistic and longitudinal studies is needed to determine whether modulating immune pathways can effectively impact disease progression.

CSF biomarkers are critical in providing insights into neurodegenerative diseases by directly reflecting biochemical and pathological changes in the brain. However, the peripheral immune system also plays a key role through immune cell infiltration, cytokine signaling, blood–brain barrier disruption, and age-related dysfunction [9]. Blood-based biomarkers offer a minimally invasive means to assess systemic immune changes. Given the bidirectional communication between peripheral and central immunity, this review examines both CSF and blood biomarker studies to compare compartmental immune responses in FTD.

Immunological biomarkers can enhance our understanding of the underlying mechanisms of FTD. They also have the potential for differential diagnosis and could be leveraged in clinical trials and patient care for early detection and disease progression monitoring. Despite extensive research on immune-related biomarkers across the innate and adaptive immune systems, a systematic review of this literature is lacking. This paper synthesizes current findings on immunological biomarkers in FTD and explores potential immune signatures to inform future research.

## 2. Methods

This systematic literature review was carried out in compliance with the Preferred Reporting Items for Systematic Reviews and Meta-Analyses (PRISMA) statement [10].

### 2.1. Eligibility Criteria

Only original, full-text articles in English were included in this review. Eligible studies quantified immunological biomarker levels in CSF or blood samples from live FTD patients and compared them to those from healthy individuals or neurological controls, including patients with any neurological disease. Comparisons among FTD subtypes were also included. Studies on markers with uncertain immunological relevance were eligible if at least three independent publications have demonstrated their direct effect on established immunological components. Postmortem and animal studies were excluded, as were reviews, case reports, scientific notes, abstracts, and duplicates.

### 2.2. Search Strategy

The PubMed and Web of Science databases were searched on 1 January 2025, using the following search string to identify studies on immune marker levels in FTD: (‘frontotemporal dementia’ OR FTD OR ‘frontotemporal lobar degeneration’ OR FTLD) AND ((blood OR serum OR plasma) OR (‘cerebrospinal fluid’ OR CSF)) AND ((immun* OR innate) OR (*inflammation OR *inflammatory) OR cytokine* OR chemokine* OR complement* OR antibod* OR microglia* OR astrocyt* OR oligodendrocyt* OR interleukin*) NOT (Review [Publication Type]). No filters were applied to the search; however, non-English studies were excluded during screening.

### 2.3. Data Extraction

Authors P.A.E., A.H.S., and A.T. designed the data extraction sheet and search strategy. The review was not registered. P.A.E. and A.T. independently screened studies by title and abstract, resolving conflicts before reviewing studies in full to assess eligibility and extract data items. Retrieved data included, where available, descriptions of FTD patients and control groups (participant numbers, age, sex, cohort matching or demographic differences, and diagnostic criteria); preanalytical variables (biofluid source, sample tube type, sampling time, storage conditions, and freeze–thaw cycles); analytical variables (analysis method, replicates, and coefficients of variation); analyzed markers (concentration and variability); and group comparison results (fold changes, *p*-values, diagnostic performance, and correlations with other parameters).

### 2.4. Data Synthesis and Analysis

Differences in biomarker concentrations across groups were assessed using fold changes and *p*-values, following the significance thresholds established by each study. Fold changes were extracted directly when available or calculated manually if group concentrations were provided. If neither fold changes nor concentrations were reported, the direction of change (increase or decrease in the FTD group versus controls) was inferred from graphical data.

Immunological biomarkers were categorized based on the direction of change, significance, and consistency across studies. Markers were classified as having the strongest evidence for a change if at least two studies reported significantly altered concentrations; evidence was considered limited if only one study reported a significant result.

Studies were synthesized based on their reported analyses. Those assessing the diagnostic performance of immunological biomarkers were included in the synthesis of diagnostic value, where reported performance measures such as area under the curve (AUC), sensitivity, and specificity were extracted when available. Studies that examined correlations between immune markers and clinical or other disease-related parameters were synthesized separately. Correlation coefficients and statistical significance were extracted. To explore the potential biological relevance of the identified immune markers, an enrichment analysis was performed to identify overrepresented pathways.

### 2.5. Enrichment Analysis

Immune markers with significantly altered levels in FTD compared to healthy con-trols were explored using the enrichment analysis tools Reactome [11], KEGG Pathways [12], and Gene Ontology (GO) [13] within the Enrichr platform [14]. Pathways containing at least five immune markers and meeting an adjusted significance threshold below 0.05 (Benjamini–Hochberg correction) were included. Unspecific immune pathways (e.g., ‘Im-mune System’ or ‘Interleukin Signaling’) were excluded.

### 2.6. Risk of Bias

The risk of bias in the included studies was assessed using a customized version of the QUADAS-2 tool [15], adapted to evaluate the methodological quality of biomarker studies rather than diagnostic accuracy (Appendix A). The tool covered four domains: (1) Patient Selection, (2) Index Test (Biomarker Measurement), (3) Reference Standard (Control Group Definition and Comparison), and (4) Flow and Timing (Sample Handling and Data Integrity). Within each domain, the risk of bias was assessed using signaling questions answered as yes, no, or unclear. These questions examined key methodological factors, including diagnostic criteria for FTD, biomarker measurement methods, control group selection, statistical approaches, and sample handling procedures. Applicability concerns were also evaluated for the Patient Selection, Index Test, and Reference Standard domains to determine whether study populations, biomarkers, and control groups were appropriate for the review’s objectives. Studies were classified as having a low risk of bias if they met all or most relevant criteria, a high risk of bias if major methodological concerns were present in one or more domains, or an unclear risk of bias if insufficient details were provided.

Since no meta-analysis was performed, we did not conduct formal sensitivity analyses or formally test for reporting bias.

## 3. Results

### 3.1. Results of Study Selection Process

The selection process yielded 1833 records from PubMed and Web of Science (Figure 1). After removing 733 duplicates, 1100 unique items remained. Screening titles and abstracts against the pre-defined eligibility criteria excluded 944 studies, leaving 156 for full-text review. Finally, 32 were excluded during the review, resulting in the inclusion of 124 studies.

### 3.2. Analytical Study Characteristics

A retrospective design was used in 115 studies (93%), and 9 were prospective (7%). CSF analysis data were reported in 74 studies (60%), while plasma, serum, and whole blood were examined in 32, 28, and 5 studies, respectively (52%). Preanalytical variables were inconsistently reported: 14 studies (11%) noted the sampling time (e.g., 10 a.m.), 50 used polypropylene tubes for biofluid storage (40%), and 86 reported storage temperatures (69%). Only 11 studies documented freeze–thaw cycles (9%). Patient and control groups were matched by sex or age in 25 studies (20%).

For analytical techniques, enzyme-linked immunosorbent assay (ELISA) was used in 71 studies (57%). Other methods included different immunoassays (17 studies), Single Molecule Array (SIMOA) (12 studies), flow cytometry (3 studies), mass spectrometry (4 studies), and proximity assays (2 studies). Quantitative PCR (qPCR), digital droplet PCR (DD-PCR), chemiluminescence, and procalcitonin (PCT) assays were each used in one study. Duplicate or triplicate analyses were specified in 50 studies (40%), and intra- and inter-assay coefficients of variation were reported in 45 (36%).

### 3.3. Biomarker Results

The 124 studies investigated 202 distinct immunological biomarkers, with 285 individual measurements, as many biomarkers were assessed across multiple biofluids. In total, 15 biomarkers were measured in at least four studies, 14 in three studies, 55 in two studies, and 201 in only one study each.

When comparing FTD patients with healthy controls (Table 1), the most substantial evidence for increased biomarker levels was found for glial fibrillary acidic protein (GFAP) in both blood [16,17,18,19,20,21,22,23,24,25] and CSF [21,26,27,28,29]. Elevated CSF levels of YKL-40 (CHI3L1) [21,29,30,31,32,33,34,35,36] and chitotriosidase-1 (CHIT1) [21,29,30,37] followed. Progranulin (PGRN) in blood showed the most evidence of decrease [38,39,40,41,42]. Additionally, four biomarkers—CCL19, CXCL1, CXCL6, and Somatostatin (SST)—were reported to have decreased levels in two studies each [43,44,45,46].

In patients with FTD, peripheral GFAP levels were consistently lower compared to those with Alzheimer’s disease (AD), as observed in plasma and serum across seven studies [16,18,19,20,24,25,27,80,100] (Appendix A). No studies reported significantly elevated blood biomarkers, and evidence for changes in CSF biomarkers was limited. While CSF YKL-40 levels were elevated in FTD compared to healthy controls, no significant differences were found between FTD and AD patients.

When comparing cohorts of FTD and amyotrophic lateral sclerosis (ALS) patients, no biomarker changes met the criteria for the ‘strongest evidence’ category, defined as being supported by two or more studies (Appendix A). One study found increased IL-15 in CSF from FTD patients [101]. In contrast, several biomarkers—including CHIT1, CXCL-12, IgG, and the ratio of neurofilament light (NfL) to YKL-40, as well as soluble APPβ to YKL-40—were significantly lower in CSF in FTD relative to ALS [36,72,102]. Additionally, circulating levels of human endogenous retrovirus group K (HERV-K) were significantly elevated in FTD [90], whereas TDP-43 antibodies, complement protein C4, and NOD2 levels were significantly decreased [96,99,102].

### 3.4. Sample Sizes and Demographics

This review did not set a minimum participant requirement per study or subgroup, acknowledging that dividing FTD by subtypes (e.g., genotype) often leads to small sample sizes. While small samples have limitations, studies on specific FTD subtypes can still provide valuable insights into disease mechanisms. Moreover, excluding studies based on sample size may introduce bias. As a result, the number of FTD patients across the included studies varied from 4 to 469, with a mean of 52.

Of the 124 studies reviewed, 25 (20%) matched cohorts for age and/or sex, while age distribution data were missing in 11 studies (9%). Correlations between age and immune biomarkers were explored in 23 studies (19%), and 8 examined the relationship between sex and these markers (6%). A summary of all tested correlations is provided in Appendix A.

### 3.5. Diagnostic Criteria for FTD

Studies were not selected based on adherence to specific diagnostic criteria, reducing selection bias while allowing the inclusion of studies that did not employ international standards. Various international diagnostic guidelines have been used to diagnose FTD, reflecting the overlapping use of phenotype-specific criteria: Rascovsky et al., 2011 [103] (74 studies, 58%), Gorno-Tempini et al., 2011 [104] (44 studies, 34%), Cairns et al., 2007 [105] (4 studies, 3%), McKhann et al., 2001 [106] (14 studies, 11%), Neary et al., 1998 [107] (30 studies, 23%), and Brun et al., 1994 [108] (2 studies, 2%). Thirteen studies (10%) did not follow international criteria. Of these, nine provided clear inclusion and exclusion criteria of their own, while four did not specify any.

### 3.6. Risk of Bias Assessment

The QUADAS-2 evaluation identified methodological concerns in several studies, primarily related to patient selection and biomarker measurement (Appendix A). Notably, the abovementioned four studies were excluded because they did not define their FTD cohort using any diagnostic criteria or clear inclusion/exclusion criteria. As a result, these studies carried a high risk of selection bias, as it could not be ensured that the included participants met established FTD diagnostic standards.

### 3.7. FTD Patients and Controls

A total of 6686 patients with FTD were included in this review, with a mean age of 63.6 years. Sex was reported in 116 studies, with 48.9% male and 51.1% female patients. FTD subtypes were specified in 52% of studies, distributed as follows: 1591 patients with bvFTD (24%), 246 with nfvPPA (4%), 215 with svPPA (3%), 31 with logopenic variant primary progressive aphasia (lvPPA, 0.4%), and 138 patients with unspecified primary progressive aphasia (PPA-NOS, 2%). Genotype data were available in 35 studies (28%), with *GRN* mutations identified in 1027 patients, followed by *C9orf72* (868 patients) and *MAPT* (443 patients).

Healthy controls (4953 participants; 47% male, 53% female; mean age 61.7 years) were included in 99 studies, while neurological controls (with any neurological condition, including other neurodegenerative diseases, subjective memory complaints, and mild cognitive impairment) were reported in 77 studies. AD patients were included in 58 studies (3758 patients: 47% male, 53% female; mean age 54.4 years), and ALS patients in 11 studies (656 patients: 52% male, 48% female; mean age 64 years).

### 3.8. Diagnostic Performance

The diagnostic performance of biomarkers in distinguishing FTD patients from other clinical groups was evaluated in 27 studies, while 62 studies examined correlations between biomarkers and other variables. Diagnostic metrics, including sensitivity, specificity, and area under the curve (AUC), were assessed for specific cut-off values of individual markers or their combinations. Analyses included comparisons between FTD and 11 studies against healthy controls [16,18,19,24,25,27,33,36,100,109,110]; 12 studies against AD [16,20,29,31,33,36,90,91,95,109,111,112]; 3 studies against ALS [72,90,95]; and 2 studies against Parkinson’s disease (PD) [27,95]. Three studies analyzed FTLD-tau versus FTLD-TDP [33,109,113], and one study investigated *GRN* mutation carriers versus non-carriers [39].

Higher plasma levels of GFAP could distinguish FTD patients from healthy controls, with AUCs ranging from 0.76 to 0.83 [16,20] (Table 2). Adding factors such as age, sex, APOEε4 status, and levels of Aβ42 and p-tau181 improved model performance to AUCs of 0.88–0.95. In contrast, increased CSF GFAP concentrations achieved an AUC of 0.71 in one study [29]. Elevated YKL-40 in CSF showed modest diagnostic value, with AUCs ranging from 0.69 to 0.88 (mean = 0.79) across five studies [29,31,33,36,109], while blood YKL-40 performed poorly, with AUCs of 0.65 against healthy controls and only 0.55 against neurological controls [91]. Similarly, other CSF biomarkers such as CHIT1 (AUC = 0.69) and MFG-E8 (AUC = 0.55) displayed limited discriminatory power [29,33]. Conversely, combinations of inflammatory miRNAs in plasma and increased HERV-K levels in FTD versus healthy controls demonstrated excellent diagnostic accuracy, with AUCs of 0.95 and 0.87, respectively [90,95].

Comparing FTD to AD, lower plasma GFAP levels produced moderate discrimination, with AUCs of 0.65–0.85 (mean = 0.78) across seven studies [16,18,19,25,27,100,110]. Adding factors such as age, sex, and APOE4 status further improved AUCs to 0.83–0.88 [16,100,112]. Elevated CSF levels of YKL-40 were poor discriminators of FTD and AD cohorts, with AUCs ranging from 0.54 to 0.6 [33,36,109]. However, one study demonstrated improved performance (AUC = 0.86) by incorporating additional variables such as age and tau levels [33].

### 3.9. Correlation of Immune Markers

The correlation analyses summarized in Table 3 include patient characteristics such as age (23 studies), sex (8 studies), disease duration (11 studies), cognitive assessments (MMSE and other tests, 11 studies each), cortical volume (8 studies), white matter lesion volume (2 studies), and various other biomarkers (34 studies), with NfL being the most frequent (6 studies).

**Table 2 biomolecules-15-00473-t002:** Reported diagnostic performances of immune markers.

	Biomarker	Specificity	Sensitivity	AUC
FTD vs. HC				
CSF	↑ GFAP	0.63 [29]	0.67 [29]	0.71 [29]
	↑ YKL-40	0.53 [109], 0.73 [29], 0.83 [31]	0.54 [31], 0.70 [29], 0.86 [109]	0.69–0.88 [29,31,33,35,109]
	↑ CHIT1	0.63 [29]	0.73 [29]	0.69 [29]
	↑ MFG-E8	NA	NA	0.55 [33]
Blood	↑ GFAP	NA	NA	0.76–0.83 [16,20]
	↑ YKL-40	NA	NA	0.65 [91]
	Infl. miRNA	0.88 [95]	0.86 [95]	0.95 [95]
	↑ HERV-K	0.67 [90]	0.92 [90]	0.87 [90]
FTD vs. AD				
CSF	↑ YKL-40	0.16 [109]	0.85 [109]	0.54–0.60 [33,35,109]
Blood	↓ GFAP	0.67 [25], 0.73 [100], 0.92 [110]	0.71 [110], 0.75 [100], 0.82 [25]	0.65–0.85 [16,18,19,25,27,100,110]
FTD vs. ALS				
CSF	↓ CXCL12	NA	NA	0.57 [72]
Blood	Infl. miRNA	0.75 [95]	0.62 [95]	0.78 [95]
	↑ HERV-K	NA	NA	0.66 [90]

The table summarizes the diagnostic performance of immunological biomarkers in differentiating patients with frontotemporal dementia from healthy controls, Alzheimer’s disease patients, or amyotrophic lateral sclerosis patients. Arrows indicate whether the biomarker levels in FTD are higher or lower compared to the control group. HC, healthy controls; NA, not available. Infl., inflammatory.

Half of the biomarkers tested for age association showed significant correlations, including GFAP, YKL-40, IL-6, IL-15, and MCP. In contrast, biomarkers like PGRN, tumor necrosis factor-alpha (TNFα), and S100 calcium-binding protein B (S100B) were not significantly related to age. Two studies reported associations between circulating TNFα and increased cerebral atrophy [82,83], particularly in *MAPT*-FTD patients, but not in those with *C9ORF72*- or *GRN*-mediated FTD [83]. Plasma tumor necrosis factor receptor 2 (TNFRII), a natural TNFα inhibitor, was also significantly correlated with greater cortical atrophy [84]. Similarly, plasma biomarkers GFAP, IL-17A, and lipopolysaccharide-binding protein (LBP) [22,82,89], as well as complement factors D and C5 [8], showed positive correlations with brain atrophy. CSF biomarkers, however, displayed distinct patterns: while CSF TNFα levels showed no association with atrophy [64], positive correlations were identified with complement proteins C1q and C3b [8]. Positive correlations with NfL were found for plasma GFAP [16,22], along with CHIT1 and YKL-40 in CSF [29]. In presymptomatic mutation carriers, but not symptomatic FTD patients, CSF levels of C1q, C3b, and plasma complement factor D were significantly associated with higher NfL levels.

### 3.10. Results of Enrichment Analysis

Enrichment analyses indicated that immune-related pathways were upregulated in blood but downregulated in the CSF of FTD patients compared to healthy controls. In blood, 43 biomarkers were elevated, while 12 were reduced. In the CSF, 28 markers increased and 44 decreased (Table 1).

Figure 2 illustrates the enriched pathways related to immune markers in CSF and blood. In CSF, enriched pathways included the complement cascade, granulocyte chemotaxis, and regulation of ERK/MAPK signaling. The complement cascade’s enrichment reflects increased activation of this innate immune pathway, which mediates synaptic pruning and neuroinflammation through microglial opsonization, aligning with the characteristic microglial activation in FTD. Granulocyte chemotaxis indicates heightened monocyte recruitment via chemotactic signals like MCP1/CCL2. ERK/MAPK signaling regulation is linked to neuroinflammation and cellular stress responses, potentially contributing to neuronal dysfunction in FTD. Pathways associated with reduced markers in CSF involved viral protein interactions, interleukin-10 signaling, and cell population proliferation. Reduced IL-10 signaling, an anti-inflammatory pathway, may impair the regulation of inflammatory cytokines like TNF-α and IFN-γ, which could accelerate disease progression. Conversely, viral protein interactions and IL-10 signaling were enriched in blood, suggesting systemic immune involvement in FTD pathology. Additionally, pathways related to leukocyte chemotaxis and IL-1 signaling were enriched in blood, supporting the idea of peripheral immune activation. Notably, no pathways were significantly reduced in blood based on the specified enrichment criteria. The few pathways containing at least five immune markers were too broad (e.g., ‘Immune System’) for meaningful interpretation. Unfiltered results are available in Appendix A.

## 4. Discussion

This review systematically evaluates existing studies that measure immunological biomarkers in biofluids from patients with FTD. A total of 124 studies were reviewed, examining the levels of 202 distinct immune markers in CSF or blood. Pathways associated with biomarkers showing significant changes were explored using enrichment analysis.

### 4.1. Methodological Variability Among Studies

This exploratory review applied broad inclusion criteria for FTD patients, resulting in considerable heterogeneity in demographics and disease subtypes. Pooling data from FTD patients against non-FTD groups risks overgeneralizing biomarker findings that may be specific to certain genotypes, phenotypes, disease stages, or demographic subgroups, such as age groups. Supporting this concern, studies indicate that glial cells display age-dependent inflammatory responses to neurodegeneration [126]. Descriptions of preanalytical variables were absent in most studies. Adherence to reporting guidelines is crucial for ensuring sample quality, reproducibility, and reliable results [127]. Only 10 studies addressed freeze–thaw cycles, which can affect biomarker measurements. For mass spectrometry on CSF samples, this effect may be negligible, as measurements remain stable after up to four freeze–thaw cycles [128]. However, certain markers like GFAP may be more sensitive [129]. Notably, few studies reported sampling time, despite circadian variations in certain inflammatory cytokines [130]. Less than half specified storage tube types and the impact of non-propylene tubes likely varies by biomarker.

### 4.2. Immunological Biomarkers in FTD vs. Healthy Controls

Numerous studies show that GFAP and MCP1/CCL2 are significantly increased in both CSF and blood in FTD patients compared to healthy controls. In CSF, CHIT1 and YKL-40 are consistently elevated as well, with some evidence for increased complement proteins C1q and C3b. Conversely, chemokines CCL19, CXCL1, and CXCL6 are reduced in CSF. In blood, IL-6, IL17A, CXCL10, and TNFα show the strongest evidence of elevation.

Increased GFAP levels in CSF and blood might indicate protein leakage into the bloodstream, potentially due to blood–brain barrier compromise. MCP1/CCL2, which modulates leukocyte inflammatory responses, has been linked to disease severity in AD and mild cognitive impairment (MCI) [131,132]. However, we observed no significant differences in MCP1/CCL2 levels between FTD and AD patients (Appendix A).

Conversely, we found suggestive evidence that the CSF levels of three chemokines—CCL19, CXCL1, and CXCL6—are significantly lower in FTD patients compared to healthy controls. CCL19 attracts B- and T-cells, while CXCL1 and CXCL6 recruit neutrophils [133,134]. This reduction may indicate dysregulated chemokine signaling in FTD, potentially impairing immune surveillance or inflammatory resolution in the central nervous system (CNS). The disparity between CSF and blood levels for the CXC chemokines suggests localized CNS-specific dysregulation, whereas the reduction in CCL19 across biofluids hints at systemic effects or reduced production, reflecting broader immune dysregulation. The precise link between peripheral and CNS alterations remains unclear and warrants further exploration.

YKL-40 is positively correlated with neuroinflammatory markers, including CSF Galectin-3, GFAP, and triggering receptor expressed on myeloid cells 2 (TREM2) [135]. Our review highlights evidence that YKL-40 levels are increased in the CSF of FTD patients but not reliably in blood, aligning with findings that blood levels remain unchanged in dementia [136]. While its role in neurodegeneration remains unclear, recent studies show no elevation of YKL-40 expression in postmortem brain samples from FTD or AD patients, raising questions about its source in these conditions [136]. Comparisons between FTD and AD reveal no significant differences in either CSF or blood, suggesting that elevated CSF YKL-40 reflects general neuroinflammation rather than a disease-specific process.

CHIT1, another chitinase expressed by activated macrophages, is elevated in the CSF across several neurodegenerative diseases, including ALS, where it promotes neuroinflammation by activating microglia and astrocytes. Astrocytes, in turn, secrete GFAP, as demonstrated in mouse models [29,137]. Our review found substantial evidence of increased CSF CHIT1 in FTD patients compared to healthy controls, suggesting its involvement in the neuroinflammatory response and potential contribution to FTD pathomechanisms. Interestingly, CHIT1 exhibits a high polymorphism frequency (1–58% depending on the population), with homozygous carriers experiencing total deficiency [138]. However, ALS patients with CHIT1 polymorphisms show no difference in disease severity [21]. Elevated CSF CHIT1 in FTD supports microglial activation, and its role in enhancing GFAP secretion aligns with the observed increase in GFAP levels in FTD.

The reviewed studies showed increases in complement proteins C1q and C3b in the CSF of FTD patients, with limited evidence supporting increases in C4 and C9. Interestingly, C1q and C4 exhibited non-significant changes in blood, suggesting local CNS production. Elevated C1q indicates classical complement pathway activation [139]. Tau conformers have been shown to significantly increase synaptic C1q in mouse models [140]. Notably, studies reporting elevated C1q involved patients with TDP-43 inclusions [17,47]—a hallmark of FTD linked to *C9ORF72* and *GRN* mutation—and one study found higher C1q in symptomatic versus presymptomatic *C9ORF72* mutation carriers [8]. This suggests that classical complement pathway activation is a broader feature of FTD, not limited to tauopathies. C3b, a convergence point in all three complement activation pathways [141], was also increased in CSF, highlighting system activation as an integral part of innate immunity in FTD. Reports of increased blood C3 raise the question of whether it originates from CNS glial cells or peripheral synthesis. However, the observation that IL-6 and IL-17A are the only cytokines significantly elevated in blood, with most inflammatory markers unchanged, challenges the hypothesis of peripheral inflammation driving elevated C3.

### 4.3. Contradictory Immunological Biomarkers

Contradictory findings were observed across studies for CX3CL1 and TNFα in CSF and CCL3 in blood. For CX3CL1 in CSF, two studies using a Proximity Extension Assay reported opposing results. Sogorb-Esteve et al. found increased CX3CL1 levels only in lvPPA patients compared to healthy controls [43], while Boström et al. observed decreased levels in an FTD cohort without subtype specification [44]. Similarly, while the lvPPA cohort exhibited significantly lower serum CCL3 levels, another study using a multiplex cytokine assay found significantly higher levels in a small bvFTD cohort [57]. These discrepancies suggest that variations in CX3CL1 and CCL3 levels may reflect subtype-specific differences in FTD, particularly in lvPPA, underscoring the importance of subtype-stratified analyses. Regarding TNFα in CSF, one study found decreased levels in *GRN*-FTD but not in sporadic FTD [50], using a Bio-Plex Pro Cytokine panel, while another study reported increased levels in an unspecified FTD cohort [64]. This indicates that TNFα changes in CSF may be associated with progranulin mutations in FTD, though methodological differences should not be overlooked.

### 4.4. CSF and Blood Discrepancies

Multiple studies have found significant upregulation of biomarkers, including IL-6, TNFα, and IL-17, in blood but not in CSF. This discrepancy may stem from the greater reliability and ease of detecting immunological biomarkers in blood, even when their concentrations are similar in both biofluids [23,50,76,81,82,84]. Alternatively, it could reflect peripheral immune activation in FTD patients or impaired CNS immune regulation due to neurodegeneration.

### 4.5. FTD Versus Alzheimer’s Disease

The comparative analysis of FTD patients against AD and ALS patients, and between FTD phenotypes and genotypes, is detailed in Appendix A.

YKL-40 levels do not differ significantly between FTD and AD patients. Consistent with prior studies, it is elevated in both diseases compared to healthy controls, as well as in ALS and prion disease, but not in vascular dementia, dementia with Lewy bodies, or Parkinson’s disease [142,143], indicating some disease specificity. Substantial evidence supports that blood GFAP levels are significantly higher in AD than in FTD [16,18,19,20,27,100,110,112], while CSF GFAP levels are not. Interestingly, beta-amyloid pathology has been proposed to be directly linked to GFAP elevation. In AD, plasma GFAP rises early in the preclinical phase, whereas in FTD and other neurodegenerative diseases, it increases during symptomatic stages [21,48,144]. The lower plasma GFAP levels in FTD compared to AD support its diagnostic utility [16], although non-immunological biomarkers like NfL and tau remain superior for distinguishing the two diseases [18].

Elevated MCP1/CCL2 levels in both CSF and blood are consistently observed in FTD compared to healthy controls but show no significant differences between FTD and AD, positioning MCP1/CCL2 as a general marker of neuroinflammation. Limited evidence suggests higher plasma protein C1 inhibitor levels in FTD compared to AD and healthy controls, despite earlier findings indicating its specificity to AD [145]. This increase may reflect a compensatory response to complement overactivation, as C1 inhibitor regulates the classical complement pathway. Finally, CSF optineurin, a regulator of macrophage function, is reduced in FTD compared to AD but elevated relative to healthy controls, indicating distinct neuroinflammatory profiles in these diseases.

### 4.6. FTD Versus ALS

FTD and ALS share genetic, neuropathological, and clinical features, with 15% of FTD patients presenting with signs of ALS and vice versa [146]. Elevated IL-15 in FTD points to increased T-cell and natural killer cell activity [101], while reduced CXCL12 may indicate impaired chemokine signaling or diminished neuroprotective cell recruitment [147]. Lower IgG levels in FTD could reflect differences in blood–brain barrier integrity or reduced autoimmune activity compared to ALS. Likewise, significantly decreased CHIT1 levels suggest less prominent microglia-mediated inflammation in FTD, which may rely more on astrocyte-linked YKL-40 expression. Distinct chitinase profiles in the ALS-FTD spectrum are supported by other emerging chitinases like CHI3L2, which may carry prognostic significance in ALS patients [148,149].

In blood, increased HERV-K DNA levels have been observed in FTD patients with TDP-43 pathology compared to ALS patients, with both groups demonstrating higher levels than healthy controls. This increase may result from TDP-43 inclusions promoting retroviral activation, though whether this drives neuroinflammation or is a consequence of TDP-43 is unclear [90].

### 4.7. Immunological Biomarkers Among FTD Subtypes

Immune markers such as GFAP, YKL-40, and CHIT1, which were consistently elevated in FTD compared to healthy controls, showed no significant differences across the major FTD phenotypes, suggesting their upregulation is a common feature of FTD. The only significantly altered biomarker was CSF Galectin-3, which was higher in bvFTD compared to svPPA and nfvPPA [32]. Galectin-3, predominantly expressed by microglia, plays a complex role in neuroinflammation, likely via the NF-KB pathway [150]. Interestingly, *MAPT*-FTD was the only genotype with significantly higher CSF Galectin-3 levels compared to *C9ORF72*- and *GRN*-mediated FTD [32], and it also displayed the lowest CSF CHIT1 levels [30]. This suggests that in *MAPT*-FTD, microglial activity is more committed to Galectin-3-mediated phagocytosis and debris clearance, possibly driven by tau accumulation, over CHIT1-mediated inflammation and matrix remodeling. Further studies are needed to explore these hypotheses.

Among the genetic subtypes of FTD, C3 was the only complement protein showing differential upregulation. Blood C3 levels were higher in *C9ORF72*- and *GRN*-FTD patients compared to sporadic FTD, suggesting its specific involvement in genetic FTD pathophysiology. This may reflect a systemic response to neuronal injury or the leakage of unconverted C3 from the CNS, where it is otherwise converted to C3b. Supporting this, C3b levels were significantly elevated in FTD patients compared to healthy controls. Other complement markers showed no differential expression across FTD genotypes. Thus, while complement activation appears to be a generalized feature of FTD, C3 may have a distinct role in its genetic forms.

### 4.8. Biomarker Correlations and Diagnostic Potential

To use biomarkers prognostically, their significant correlation with disease burden is essential. Immunological markers like CSF C1q and C3b [8], CHIT1 [29], YKL-40 [29], and plasma GFAP [16,22] show positive correlations with NfL levels (Table 2). This association may result from either neurodegeneration-induced immune activation or pathogenic immune activation causing neuroinflammation, which elevates NfL levels. Interestingly, one study found that CSF C1q and C3b correlated with NfL levels only in presymptomatic participants, also when adjusting for age, suggesting that complement activation occurs in the late preclinical stage alongside early brain atrophy [8].

Most age-correlated biomarkers were pro-inflammatory, except for transforming growth factor beta 1 (TGF-β) in CSF and secretory leukocyte peptidase inhibitor (SLPI) in blood. This could indicate that the extent of neuroinflammation increases with age in FTD patients. However, the potential influence of nonspecific, age-related baseline inflammation (‘inflammaging’) cannot be overlooked, underscoring the importance of additional age-matched studies [151].

Based on our diagnostic performance data, a combination of plasma GFAP and CSF YKL-40 most effectively differentiates FTD patients from controls. Our review also highlights the potential of using several immunological proteins, particularly complement factors from the classical pathway, for disease specificity. Moreover, elevated plasma GFAP and increased CSF levels of CHIT1, C1q, and C3b may help distinguish FTD from AD.

### 4.9. Enriched Immunological Pathways in FTD

This analysis revealed an overall enrichment of immune-related pathways in blood and a reduction in CSF. This trend may reflect a loss of immune regulation in the CNS due to neurodegeneration, a compensatory upregulation of peripheral immune responses, or the leakage of immunological proteins from the CSF into the bloodstream, as observed in certain FTD cohorts [152].

Notably, the enrichment analysis highlights compartment-specific immune mechanisms in CSF and blood. Increased blood markers in the IL-10 pathway suggest systemic inflammation with compensatory IL-10 upregulation, while reduced CSF markers indicate diminished CNS anti-inflammatory and neuroprotective activity, potentially worsening neuroinflammation and disease progression. These findings suggest that therapeutic strategies should target the distinct immune environments, enhancing IL-10 signaling in the CNS for neuroprotection while mitigating systemic inflammation without disrupting IL-10 activity in the blood.

Several overrepresented pathways are related to the chemotaxis of peripheral immune cells, which are recruited to the CNS in neurodegenerative diseases like multiple sclerosis (MS), AD, ALS, and PD [153,154]. In the EAE model of MS, neutrophils recruited to the CNS show increased IL-10 expression, promoting neuroprotection [153]. The enriched IL-10 pathway in this analysis may intersect with pathways related to immune cell recruitment. The simultaneous enrichment and reduction in granulocyte chemotaxis markers in CSF suggest nuanced regulation, impaired neutrophil recruitment, or altered function, contributing to the complex immune dysregulation in FTD and the interplay between central and peripheral immunity.

### 4.10. Limitations

The database searches in this study were conducted without filters, as options like ‘English’ or ‘Human’ might exclude relevant misclassified studies. However, non-English studies were later excluded during screening, which may have introduced bias.

Given our explorative approach, where biomarkers were ranked by the number of studies reporting significant alterations, selection bias could overestimate the strength of evidence for biomarkers disproportionately represented in studies with significant outcomes.

The classification of biomarkers as ‘immunological’ may lead to bias due to the lack of a clear definition in the literature. In this study, we included biomarkers without well-documented immune functions if at least three independent studies demonstrated direct interaction with established immune components.

Our inclusion criteria did not mandate specific diagnostic criteria for FTD, which broadened the pool of studies but also increased cohort heterogeneity. However, earlier diagnostic guidelines (e.g., Neary et al., 1998 [107]) were clinically more conservative than recent criteria (Rascovsky et al., 2011 [103]), which introduced the category of ‘possible FTD’ to enhance sensitivity and support earlier diagnosis.

To maintain a broad overview, we did not differentiate between plasma, serum, and whole-blood measurements, though these biofluids have distinct compositions that can affect biomarker levels. For example, serum may show higher concentrations due to platelet activation during clotting, while plasma may provide more accurate measurements by avoiding clotting-related substances. Whole blood may release biomarkers through lysis during sample handling.

The lack of age or sex matching in 80% of the studies reviewed presents a confounding factor, as immunosenescence and hormonal differences can influence immune activity [155]. Additionally, comorbidities, particularly inflammatory diseases, were rarely reported and therefore not extracted separately. Only 7% of studies were prospective, limiting our ability to assess how immunological biomarkers change over time in FTD. Future research should prioritize longitudinal designs to track biomarker dynamics, establish causal links, minimize recall bias, and provide standardized data on disease progression. Identifying biomarkers linked to disease stages or treatment responses could inform clinical decision-making, improve patient monitoring, and help refine potential therapeutic approaches.

## 5. Conclusions

This review provides an overview of immunological biomarkers in sporadic and genetic FTD. Comparisons with healthy controls identified GFAP and MCP1/CCL2 as prominent immune markers in both CSF and blood, with CHIT1 and YKL-40 consistently elevated in CSF. Complement proteins, especially from the classical activation pathway, are promising targets for further investigation. When comparing FTD and AD cohorts, GFAP, SPARC, and SPP1 emerged as relevant markers, while IL-15, HERV-K, NOD2, and CHIT1 stood out in comparison with ALS. Among FTD phenotypes, Galectin-3 was the only significantly altered biomarker in bvFTD compared to PPA, and it was significantly higher in *MAPT*-FTD than in other genotypes, potentially reflecting interactions between microglia and tau pathology. Enrichment analysis pointed to IL-10 signaling and peripheral immune cell recruitment, underscoring compartment-specific immune regulation in CSF and blood in FTD.

This review also highlights the significant variability in study methodologies and cohort compositions, stressing the need for standardized approaches, larger and more diverse study populations, and prospective longitudinal studies. Such efforts are crucial to enhance reproducibility and generalizability and to better understand immune marker dynamics and their correlation with disease progression. Despite progress, many questions remain regarding the immune system’s role in FTD pathology, and this review identifies key areas for future research.

## Figures and Tables

**Figure 1 biomolecules-15-00473-f001:**
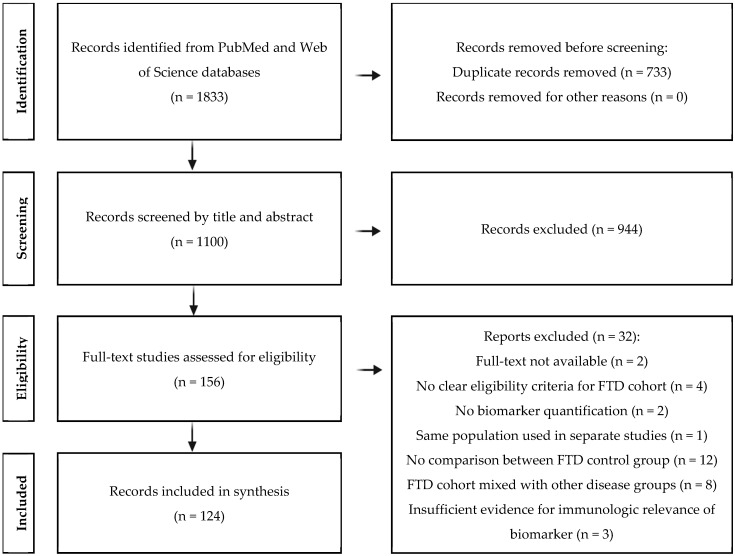
A PRISMA flow diagram illustrating the literature selection process.

**Figure 2 biomolecules-15-00473-f002:**
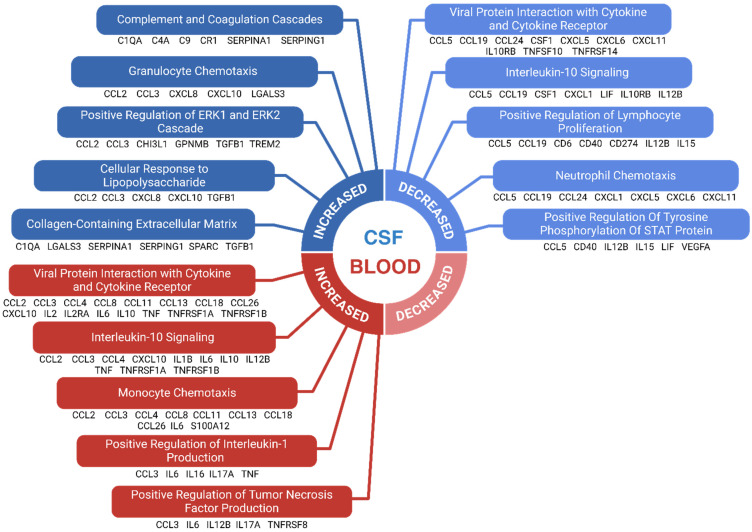
Enriched immunological pathways in FTD. The enrichment analysis was conducted on immunological markers with significantly altered levels in FTD patients compared to healthy controls. Each selected pathway required a minimum of 5 biomarkers and had to meet an adjusted significance threshold of 0.05. Broad, non-specific immune pathways were excluded. The top 5 pathways, ranked by adjusted significance levels, are shown for immune markers with increased and decreased levels in CSF, as well as increased levels in blood. No pathways were identified for markers with reduced levels in blood when using the stated selection criteria. Biomarkers belonging to each enriched pathway are shown below it.

**Table 1 biomolecules-15-00473-t001:** Comparative analysis of immune markers in FTD patients and healthy controls.

Reported Change in CSF Levels	FTD Versus Healthy controls CSF-Based Immune Markers
↑↑	C1q [8,17,47] C3b [8,17] CHIT1 [21,29,30,37] GFAP [18,21,26,27,28,29,48] MCP1/CCL2 [43,49,50,51,52] YKL-40 [21,29,30,31,32,33,34,35,36,37,53,54,55,56]
↑	β-NGF [44,57] C4 [58] C9 [59] CCL3 [43,49,51,52] Gal-3 [32] GPNMB [60] IL-8 [45,50,51,61] Leukotrienes [62] MMP-10 [44] NfL/YKL-40 ratio [35] SERPINA1 [63] SERPING1 [56] SPARC [56] TGF-β [52,64] TREM2 [37,65,66] Thromboxane [62] CD56+ [67] CCR7- CD45RO+ CD4+ [67] CCR7-CD45RA+ CD8+ [67]
↓↓	CCL19 [43,44] CXCL1 [43,44] CXCL6 [43,44] SST [44,45]
↓	ADA [44] Aβ42/YKL-40 ratio [34] CCL24 [44] CD5 [44] CD6 [44] CD8A [44] CD40 [44] CD244 [44] CDCP1 [44] CSF-1 [44] CXCL5 [43] CXCL11 [43,44] Cystatin C [56] DNER [44] FGF-5 [44] FGF-19 [44] Flt3L [44] HGF [44,52] IL-10RB [44] IL-12B [44] IL-15 [49,50,61] LIF [44] MCP-2 [44] PD-L1 [44] PGRN [68] RANTES [50] sAPPβ/YKL-40 ratio [35] SCF [44] TGF-α [44] TNFRSF9 [44] TNFRSF14 [44] TRAIL [44] TWEAK [44] uPA [44] VEGF [44,49,50] CD45RA- CD45RO+ CD4+ [67] CCR7+ CD45RO+ CD4+ [67] CCR7+ CD45RO+ CD8+ [67] CCR7- CD45RO+ CD8+ [67]
→→	CCL4 [43,49,50] CCL11 [43,49] CXCL10 [43,49,51] CCL11 [43,50] FGF-2 [50,57,61] G-CSF [50,61] GM-CSF [50,61] IFN-γ [49,50,61] IL-1RA [50,61] IL-2 [50,61] IL-4 [50,61] IL-5 [49,50,61] IL-6 [50,61] IL-7 [49,50,61] IL-9 [50,61] IL-10 [49,50,61] IL-12 [49,50,61] IL-13 [50,61] IL-17 [50,61] Leukocytes [69,70]
→	C8G [71] CCL8 [43] CCL13 [49] CCL17 [49] CCL22 [49] CCL23 [43] CCL25 [43] CCL26 [49] CCL28 [43] CFB [47] CXCL9 [43] CXCL12 [72] DKK-3 [73] GFAP CSF/serum ratio [27] ICAM-1 [74] IL-1β [50] IL-16 [49] KYNA [75] MFG-E8 [32] PDGF-BB [50] Prostaglandins [62] VCAM [74] VEGF [50] CD14+ CD3- [67] CD14+ CD3+ [67] CD56+ CD3+ [67] HLA-DR CD14+ [67] CD3+ CD14- [67] CD4+ [67] CD8+ [67] CD19+ [67] HLA-DR CD3+ [67] CD69+ CD3+ [67] CD45RA- CD45RO- CD4+ [67] CCR7+ CD45RA+ CD8+ [67] CD45RA+ CD45RO- CD4+ [67] CD45RA+ CD45RO+ CD4+ [67] CD45RA- CD45RO- CD8+ [67] CD45RA+ CD45RO- CD8+ [67] CD45RA- CD45RO+ CD8+ [67] CD45RA+ CD45RO+ CD8+ [67] CCR7+ CD45RA+ CD4+ [67] CCR7- CD45RA+ CD4+ [67]
Contradictory	CX3CL1 [43,44] TNFα [49,50,52,64]
Reported change in blood levels	Blood-based immune markers
↑↑	C3 [8,76,77,78,79] CXCL10 [23,43,50,51,57] GFAP [16,17,18,19,20,21,22,23,24,25,27,80] IL-6 [49,57,76,77,81] IL-17A [23,57,82] IL-1β [23,38,50,57,77,82] MCP1/CCL2 [23,43,51,57] TNFα [23,50,57,77,81,82,83,84,85,86]
↑	ANA [87] APOL1 [78] APN [88] BAFF [82] C2 [8] C3 in EVs/plasma [79] CD163 [89] CCL4 [23,43,49,50] CCL8 [43] CCL11 [23,50,57] CCL13 [43,49] CCL18 [89] CCL26 [23] HERV-K [90] IL-2 [49,50,57,61,82] IL-2RA [57] IL-6 [23,50,82] IL-7 [23] IL-10 [23,49,50,57,77,82] IL-12p70 [23,49,57,77,82] IL-15 [23,49,50,57] IL-16 [23,49,57] MCP-4 [23] Myosin-2 [78] S100A8 [78] S100A12 [38] SCGF-BB [57] SLPI [42] TNFα [23] TNFRI [84] TNFRII [84] TNFRSF8/CD30 [82] TWEAK [82] YKL-40 [16,83,91]
↓↓	PGRN [38,39,40,41,42,50,92]
↓	C1s [78] C7 [49,50,57,78] CCL19 [43] CCL20 [43] GSN [78] IL-1α [23,49,57] IL-5 [23] IGHV1-2 [78] PROS-1 [78]
→→	C1q [8,79] C4 [8,79] CCL11 [43,49] CCL23 [43,49] CRP [77,89,93] FGF-2 [50,57,61] G-CSF [50,57] GM-CSF [23,50,57] IFN-γ [23,49,50,57,77,82] IL-1RA [50,57] IL-4 [23,50,57,77,82] IL-5 [49,50,57,77] IL-8 [23,49,51,57,77] IL-9 [50,57] IL-12 [23,49,50] IL-13 [23,49,50,57,61] IL-17 [49,50,57] IL-18 [49,57,89] miRNA [94,95] VEGF [23,49,50] TNFβ [23,57]
→	Anti-β2GPIs [87] Anti-cardiolipin [87] Anti-TDP43 [96] BACE-1 [24] Basophils [93] BP180 NC16A [97] BP230 [97] C4B [8] C5 [8] Cathepsin S [98] CCL7 [43] CCL17 [49] CCL22 [49] CCL25 [43] CCL26 [49] CCL28 [43] CD14 [89] CFB [8] CFI [8] CFH [8,77] CTACK [57] CX3CL1 [43] CXCL1 [43] CXCL5 [43] CXCL6 [43] CXCL9 [43] CXCL11 [43] Gal-3 [32] GRO-α [57] HGF [57] IFN-α2 [57] IL-12p40 [57] IL-1 [86] IL-2 [23] IL-3 [57] IL-22 [77] Irisin [81] LBP [89] LIF [57] MDC [23] MIF [57] MIG [57] NOD2 [99] PDGF-BB [50,57] RANTES [50,57] SCF [57] SDF-1α [57] SPP1 [99] TARC [23] TRAIL [57] TREM2 [86] TWEAK [84]
Contradictory	CCL3 [23,43,49,50,57]

The table presents immune markers quantified in the CSF and blood of FTD patients compared to healthy controls. Markers are categorized based on the direction of change and the strength of supporting evidence. Two vertical arrows indicate at least two studies reporting significantly elevated or reduced levels of FTD, while a single vertical arrow represents one study with significant findings. Biomarkers for which one or more studies have shown unchanged levels are marked with one or two horizontal arrows, respectively. ‘Contradictory’ indicates reports of both significantly increased and decreased levels. ‘Blood levels’ refers to measurements in serum, plasma, or, in a few cases, whole blood. Appendix A provides the references for all biomarker results.

**Table 3 biomolecules-15-00473-t003:** Reported correlations between immune markers and other variables.

	Age	Sex	NFL	Cortical Atrophy	MMSE
Positive Correlations					
CSF	CXCL9 * [43] IL-15 * [49,50,101] MCP1/CCL2 ** [50,61,114] TREM2 * [49] TGF-β * [64] YKL-40 * [115]		C1q (presymp- tomatic) *** [8] C3b (presymp- tomatic) *** [8] CHIT1 *** [29] YKL-40 *** [29]	C1q (presymp- tomatic) ** [8] C3b (presymp- tomatic) ** [8]	
Blood	CCL13 * [49] GFAP *** [18,25] IL-1β * [38] IL-6 *** [41,51,83] IL-8 * [49] SLPI *** [42] TNFα *** [83] YKL-40 *** [83,91]	GFAP (females) *** [25]	Complement Factor D (presymptomatic) *** [8] GFAP *** [16,22]	C5 * [8] Complement Factor D *** [8] GFAP * [22] IL-17A * [82] LBP * [89] TNFα * [82,83] TNFRII ** [84]	
Negative correlations					
CSF		CCL28 * (females) [43]			CHIT1 (presymptomatic *GRN*-FTD) ** [37]
Blood					GFAP ** [16,25] MCP1/CCL2 * [116]
No correlation					
CSF	GFAP [18] IL-11 [117] IL-15 [101] CXCL10 [50] PGRN [118] RANTES [50] S100B [119] SORL1 [120] SST [45] TNF [50]	CCL2 [43], CCL3 [43] CCL4 [43] CCL8 [43] CCL11 [43], CCL19 [43] CC23 [43], CCL25 [43] CCL28 [43], CX3CL1 [43] CXCL1 [43], CXCL5 [43] CXCL6 [43], CXCL9 [43] CXCL10 [43], CXCL11 [43], GFAP [38] IgM and IgG gangliosides [121] IL-6 [117], IL-8 [43] IL-11 [117], IL-15 [101] SST [45]		TNFα [64] TGF-β [64] IL-1β [64] S100B [119] Monocytes [122]	CHIT1 [37] GFAP [28] IL-15 [101] IgM and IgG gangliosides [121] S100B [119] TREM2 [37,65] YKL-40 [37]
Blood	Cathepsin S [98] PGRN [38,41,42,118,123,124]	Cathepsin S [98] GFAP [18]	C9 [59] Gal-3 [32]	BAFF [82], IFN-y [82] IL-1b [82], IL-2 [82] IL-4 [82], IL-6 [82,83] (*GRN*-FTD) IL-10 [82] IL-12p70 [82] sCD30 [82] TWEAK [82] YKL-40 [82]	IL-6 [125] IL-18 [125] TNF [84] TNFRI [84] TNFRII [84] TWEAK [84]

The table summarizes correlations between immune markers and the most frequently assessed variables in the included studies. Results are categorized as significantly positive, significantly negative, or non-significant. * *p* < 0.05, ** *p* < 0.01, *** *p* < 0.001.

## Data Availability

The raw data supporting the conclusions of this article will be made available by the authors on request.

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
