# Peer review of "Immunological Fluid Biomarkers in Frontotemporal Dementia: A Systematic Review"

_biomolecules, 2025, doi:10.3390/biom15040473_

Round 1
Reviewer 1 Report
Comments and Suggestions for Authors
1) I would recommend expanding the introduction
2) The article reads like a set of abstracts, excessively divided into subsections. The authors should rewrite the presented results.
3) The conclusion contains an excessive number of abbreviations
4) The tables provide references, however, in many cases these are references to the same work. A more detailed analysis of the literature should be presented.
Reviewer 2 Report
Comments and Suggestions for Authors
In this systematic review, Erichsen et al. follow the guidelines set forth in the PRISMA statement to summarize the current findings regarding the potential of immune-related proteins or immune profiles as biomarkers for frontotemporal dementia (FTD) in an effort to provide a resource to guide future studies. The authors used a Boolean search string to query both the Web of Science and PubMed databases for source material. After filtering for duplicate records, relevance, and compliance with the described eligibility criteria the authors included 128 original reports on immunological protein levels in the blood and/or CSF collected from living FTD patients, patients suffering from other neurological diseases (neurological controls), or healthy control individuals. The authors then succinctly summarize the results, sorting putative biomarker proteins by the number unique reports describing altered or equivalent expression levels relative to control groups. Importantly, the authors also summarize correlations between the immune proteins of interest with covariates such as age and sex, as well as the directionality of those correlations. Finally, the authors performed enrichment analysis to describe the specific immune pathways and the immunological proteins within them that are altered in the CSF and blood of FTD patients. The current manuscript highlights:1) the diagnostic performance of individual proteins or sets of protein in distinguishing FTD patients from other groups; 2) Immune-related proteins that have been described to change in the CSF or plasma of FTD patients relative to a variety of control groups, particularly those that have been described in multiple publications; 3) Correlations between immunological proteins and variables other than disease that should be considered when evaluating putative biomarkers; and 4) specific immune pathways altered in FTD, and the directionality of change in the two biofluids analyzed.
As it stands, the current article will provide a useful resource for those working to find diagnostic and/or prognostic biomarkers for frontotemporal dementia, as well as other related diseases such as Alzheimer’s and amyotrophic lateral sclerosis. In addition, the discussion of pre-analytical factors that can influence biomarker studies is important as the field strives to increase the reproducibility of results coming from an international research community. However, there are some changes I feel could improve the manuscript overall and have included suggestions to that end.
- Text in the bottom right panel of figure 1, describing studies that did not meet eligibility criteria is cut off/doesn’t fit in the box.
- When discussing comparisons of immunological biomarkers between FTD and ALS cohorts, the authors state that “no biomarkers showed strong evidence of significant differences in CSF or blood” (lines 153-154) but then go on to discuss several reports of significant decreases in biomarkers. Did the quoted sentence mean to say “significant increases” rather than “differences”? This small paragraph (Lines 153-159) should be edited for clarity.
- The authors summarize significant alterations in protein levels for putative biomarkers, and correlations with covariates other than disease. A table summarizing the diagnostic performance of biomarkers discussed in 3.7 would improve the manuscript.
- Can you provide a short justification for including results of the eight studies that provided no explanation regarding inclusion/exclusion criteria in their study?
- Sometimes when mentioning the number of studies that following certain criteria (e.g., age and sex matching participants), a percentage is also reported, but not always. It could help to make this uniform throughout the paper.
- While the unfiltered enrichment analyses are available for review as supplementary figures, a few words on which enrichment analysis criteria were failed would be helpful for the reader when describing why there are no pathways significantly reduced in blood (paragraph on lines 244-249).
Reviewer 3 Report
Comments and Suggestions for Authors
This is a very valuable review manuscript that addresses clear unmet medical demand.
1. Can the authors provide more information on the relevance of using blood as a biofluid to study the brain? Maybe they can add a paragraph under the discussion. This is a highly relevant issue that needs to be addressed.
2. The same goes with the CSF. Does CSF reflect the processes from the brain or the blood, considering the origin of the CSF?
3. The comparison of FTD with other diseases is very relevant and there are many commonalities between different neurodegenerative diseases. The authors have FTD versus ALS part, but maybe they can extend this a little and consider these recent papers (PMID: 39844875, 38129934, 38001563). These papers seemed to be left out, but these papers describe the CHI3L2 as one potential biomarker and some other overlapping genes.
Reviewer 4 Report
Comments and Suggestions for Authors
Thank you for the invitation to review this work by Erichsen and colleagues. The systematic review explores into immunological biomarkers for FTD. I have several comments for authors to consider
- The introduction discusses neuroinflammation's role in FTD but does not adequately differentiate between the primary and secondary roles of immune dysfunction in disease progression. Authors may further clarify the link (association/causality) between immune markers and disease mechanisms.
- While the authors acknwledge the lack of disease-specific biomarkers, the introduction could expand on how immunological biomarkers might fill this gap by providing perspectives on potential of immunological biomarkers to differentiate FTD from other neurodegenerative diseases.
- Methods, the search string excludes studies in non-English languages, which may introduce bias. This should be acknowledged in the limitations.
- The irsk of bias assessment was qualitative, suggest QUADAS-2 may be used.
- Table 2 does not clearly indicate the statistical significance (e.g., p-values) of reported changes. Add a column to the table to indicate the significance of findings for each biomarker.
- Heterogenity results highlight inconsistencies in biomarker levels (e.g., TNFα, CX3CL1) but do not explore the potential reasons for these discrepancies (e.g., analytical methods, cohort characteristics). Please discuss potential source of the heterogenity.
- In Page 10, line 20, suggest adding a more detailed interpretation of enriched pathways (e.g., IL-10 signaling, complement cascade) and their implications for FTD.
- Authors should add the need for prospective studies to assess temporal changes in biomarkers and their linkages to disease progress in the Discussion.
- In Table 1, provide a reference for each included biomarker.
- The conclusion could emphasize the need for standardized methodologies and larger, more diverse cohorts.
Round 2
Reviewer 1 Report
Comments and Suggestions for Authors
The authors took into account all my comments. The quality of the article has been significantly improved. The question of a reference in the table to one work is justified. I believe that the article can be accepted in its current form.
Reviewer 2 Report
Comments and Suggestions for Authors
All concerns were addressed.
Reviewer 4 Report
Comments and Suggestions for Authors
No further comments